# Ultrafast imaging of polariton propagation and interactions

Ding Xu[1,3], Arkajit Mandal [1,3], James M. Baxter[1], Shan-Wen Cheng[1], Inki Lee[1], Haowen Su [1], Song Liu[2], David R. Reichman [1] & Milan Delor [1]

Semiconductor excitations can hybridize with cavity photons to form exciton-polaritons (EPs) with remarkable properties, including light-like energy flow combined with matter-like interactions. To fully harness these properties, EPs must retain ballistic, coherent transport despite matter-mediated interactions with lattice phonons. Here we develop a nonlinear momentum-resolved optical approach that directly images EPs in real space on femtosecond scales in a range of polaritonic architectures. We focus our analysis on EP propagation in layered halide perovskite microcavities. We reveal that EP–phonon interactions lead to a large renormalization of EP velocities at high excitonic fractions at room temperature. Despite these strong EP–phonon interactions, ballistic transport is maintained for up to half-exciton EPs, in agreement with quantum simulations of dynamic disorder shielding through light-matter hybridization. Above 50% excitonic character, rapid decoherence leads to diffusive transport. Our work provides a general framework to precisely balance EP coherence, velocity, and nonlinear interactions.

Exciton-polaritons (EPs) form when semiconductor excitons (electron-hole pairs) hybridize with photons, resulting in renormalization of light-matter eigenstates[1,2]. Hybridization is readily attained with or without photonic cavities at room temperature in two-dimensional (2D), hybrid and molecular semiconductors that sustain strong light-matter interactions and large exciton binding energies[3–6], paving the way to scalable polaritonic devices. EPs exhibit highly desirable properties. Photon-like EPs are highly coherent and exhibit long-range ballistic energy flow ideal for energy technologies[7–10]. Exciton-like EPs sustain ultra-strong nonlinear interactions that could lead to single-photon quantum switches[11–14]. Nevertheless, the true promise of EPs emerges when their light-like and matter-like features are combined. In this regime, many open questions remain: can EPs with high exciton character (>50%) preserve their intrinsic group velocities even in the presence of exciton-mediated scattering with phonons and other species? At what exciton fraction do matter-like interactions lead to EP decoherence? Maintaining long-range ballistic transport for highly matter-like excitations would enable, for example, large-scale photonic circuits based on single-photon quantum gates. Simultaneously optimizing EP transport and nonlinearities for any given system requires new high-throughput approaches capable of directly tracking EP propagation and interactions throughout their lifetimes.

Fast propagation and nonlinear interactions make EPs exceptionally challenging to study on their intrinsic spatiotemporal scales. Several powerful approaches have been developed to image EPs, though the majority rely on steady-state far-field optical microscopies[10,15–17] that are unable to directly track EP propagation and nonequilibrium processes. Some ultrafast implementations of non-linear far-field microscopies have been applied to polaritonic systems[18–20], but importantly lack the momentum resolution or specie-specificity to probe EPs and their interactions with other material excitations. Other powerful implementations based on detecting polariton or condensate emission in the far field using streak cameras[21,22] or interferometry[23–25] enable real-space monitoring of polariton evolution, but require slow-moving, long-lived and brightly emissive species as well as high excitation fluences, limiting the generalizability of these approaches and the achievable signal-to-noise ratios in delicate measurements. Recent tour-de-force experiments

[1]Department of Chemistry, Columbia University, New York, NY 10027, US. [2]Department of Mechanical Engineering, Columbia University, New York, NY 10027, US. [3]These authors contributed equally: Ding Xu, Arkajit Mandal. ✉e-mail: drr2103@columbia.edu; milan.delor@columbia.edu

using ultrafast near-field scanning[26–28] and electron microscopies[29] have been applied to non-cavity EPs and phonon-polaritons in van der Waals materials, boasting high momentum and spatial resolution, allowing direct tracking of polariton wavepackets on sub-picosecond scales. Nevertheless, near-field approaches are not generalizable to microcavity EPs and other complex material architectures that require sub-surface penetration, and the extension of electron microscopies to microcavity EPs and fragile materials remains untested.

Here we develop a highly generalizable and noninvasive approach based on spatiotemporally resolved far-field optical microscopy[30,31] allowing direct imaging of EP propagation and interactions on femtosecond–nanosecond scales with sub-100 nm spatial sensitivity. We term our approach Momentum-resolved Ultrafast Polariton Imaging (MUPI). By directly tracking EPs and excitons throughout their lifetimes with high spatiotemporal precision, we quantify key factors affecting EP propagation, including EP–lattice and EP–EP scattering as a function of light versus matter composition, quantities that were never directly empirically accessed. We demonstrate that our approach is generalizable across microcavity EPs, self-hybridized EPs in material slabs without external cavities, and plasmon-exciton (plexciton) polaritons in a range of emerging molecular and material systems. We focus our analysis on 2D halide perovskite microcavities at room temperature, an ideal test system that reaches the strong coupling limit without complicated cavity fabrication, possesses strong EP interactions[32], and displays low intrinsic exciton diffusivity[33,34] allowing unambiguous spatial isolation of excitonic versus EP signals. We reveal that EPs with large exciton character in these systems are strongly affected by scattering with lattice phonons, leading to up to 40% renormalization of EP velocity at room tem-perature. Remarkably, despite these strong EP–phonon interactions, EPs retain ballistic transport properties for up to 50% exciton char-acter. Beyond 50% exciton character, matter-mediated interactions lead to decoherence and diffusive transport in our structures.

## Results

### MUPI imaging of diffusive exciton and ballistic EP propagation

MUPI is illustrated in Fig. 1a (see Figs. S1–S2 for more detail). A diffraction-limited femtosecond visible pump pulse generates exci-tons or EPs by exciting the material either above-gap or at a polariton resonance. A widefield backscattering probe[35] then images the sample with and without the pump pulse at controlled time delays. Differential pump ON/pump OFF images provide a direct readout of the spatial distribution of pump-generated species, which can be tracked with sub-diffraction spatial precision[30,31] (Supplementary Note 1). Momentum-matched probing of different EP modes (Fig. 1b) in MUPI is achieved by displacing the probe along the optical axis of the objec-tive, resulting in tilted widefield illumination of the sample[36]. The use of a high numerical aperture microscope objective (NA = 1.4) provides access to EP momenta $k = 2\pi \, NA/\lambda$ greater than 10 $\mu m^{-1}$, where $\lambda$ is the wavelength of light. The sample can be imaged either in real space to track exciton and EP propagation (Fig. 1c–g), or in momentum-space to provide the linear and nonlinear (excited state) EP dispersion through angle-resolved reflectance spectra (Fig. 1b, h, i).

We focus our analysis on a 0.67 μm thick slab of the layered halide perovskite $(CH_3(CH_2)_3NH_3)_2(CH_3NH_3)Pb_2I_7$ flanked by two metallic mirrors (Fig. S3a). The dispersion of our structure is shown in Fig. 1b, with three resolved lower polariton (LP) branches. A fit to the experi-mental dispersion using a coupled oscillator model indicates a Rabi

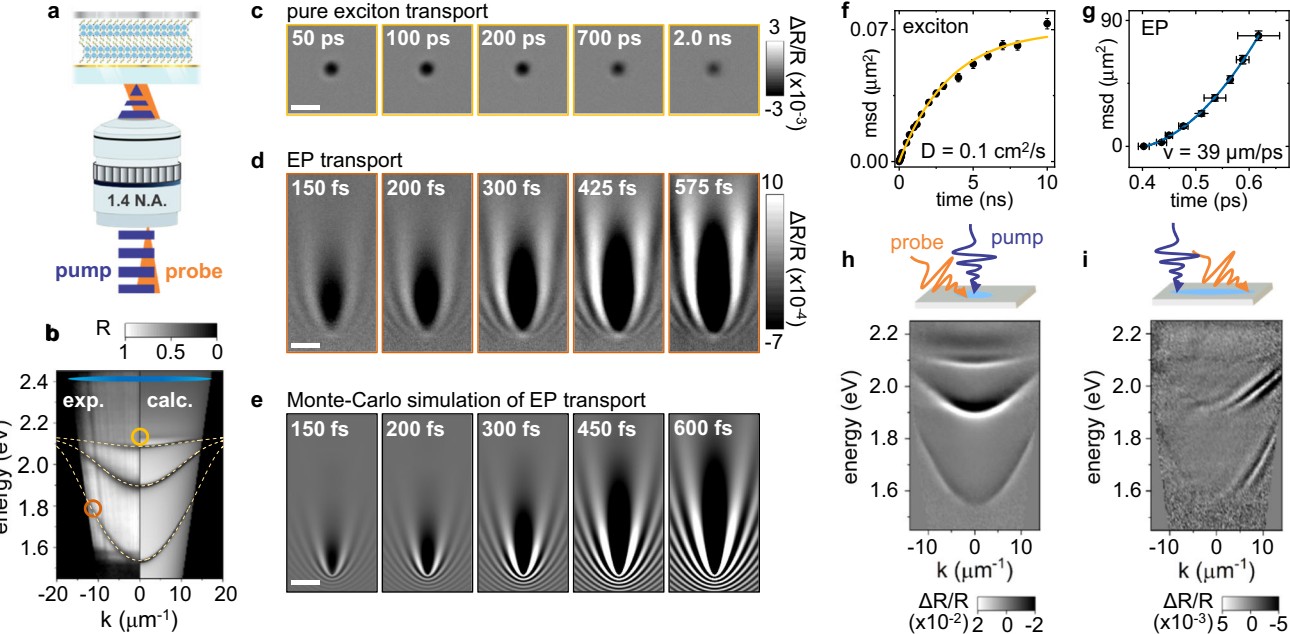

**Fig. 1 | Tracking EPs in a layered halide perovskite microcavity. a** MUPI setup and sample. **b** Momentum-resolved white light reflectance spectrum of our primary sample, a 0.67 μm thick slab of the layered halide perovskite $(CH_3(CH_2)_3NH_3)_2(CH_3NH_3)Pb_2I_7$ flanked by two metallic mirrors. Dashed lines cor-respond to a coupled oscillator model fit; the right side of the figure is a scattering matrix simulation of the structure (Fig. S4). **c** Exciton transport probed at $k = 0$, $E = 2.17$ eV following above-gap (2.41 eV) pump excitation. **d** EP transport probed at $k = 8.98 \, \mu m^{-1}$, $E = 1.77$ eV following 2.41 eV pump excitation. The probe energies and momenta used for panels (**c**) and (**d**) are illustrated with circles in panel (**b**); the pump energy and momentum range is illustrated with a blue ellipse. **e** Monte-Carlo simulation of MUPI contrast generated during EP propagation. Scale bars for panels (**c**–**e**) are 2 μm. **f** Mean squared displacement (msd) of bare excitons from data in panel (**c**). Error bars are one standard deviation. The solid curve is a fit assuming trap-limited diffusive transport. **g** msd of EPs from data in panel (**d**). Error bars are one standard deviation. The solid curve is a fit assuming ballistic transport.
**h** Differential pump ON/pump OFF angle-resolved reflectance spectrum obtained at 1 picosecond pump-probe time delay, displaying pump-induced modification to the EP dispersion when pump and probe beams are spatially overlapped. **i** Same as panel (**h**), with the probe spatially separated from the pump by 1.1 μm, selectively probing EP species that have propagated away from the excitation spot.

splitting of 275 meV (Fig. S4), similar to previous reports[32]. Figure 1c displays excitonic transport in this structure by pumping above-gap and probing at resonance with the exciton reservoir at $k = 0$ (see also Supplementary Movie 1 and analysis in Fig. S6; the probe energy and momentum are highlighted with the yellow circle in Fig. 1b). Figure 1f plots the exciton mean squared displacement (msd), defined as $msd = \sigma^2(t) - \sigma^2(0)$, where $\sigma$ is the Gaussian width of the population profile, and $t$ is the pump-probe time delay (Supplementary Note 1). The exciton msd is exponentially-decaying over a few nanoseconds, characteristic of trap-limited diffusive transport[34]. We extract a diffusivity of $0.10 \pm 0.03$ cm$^2$/s and a trap density of 29 µm$^{-2}$. These values are in good agreement with recent reports of cavity-free exciton transport in these materials[33,34], suggesting that the transport of bare excitons is unaffected by the cavity.

Figure 1d displays EP propagation at the same location using the same above-gap pump excitation conditions, but probing at an energy and momentum corresponding to the orange circle in Fig. 1b (see also Supplementary Movie 2). The MUPI image series in Fig. 1d displays a fast-propagating EP signal that extends over several microns within a few hundred femtoseconds, in stark contrast to the practically static bare exciton signal of Fig. 1c. The intensity of the EP signal only becomes substantial ~300 fs after non-resonant pump excitation, since the latter primarily populates excitons uncoupled to the cavity; these reservoir excitons scatter into the LP branch on timescales dictated by exciton–phonon scattering[37,38], leading to a delayed rise of the EP signal. The EP msd in Fig. 1g is extracted by spatially tracking the EP wavefront (analysis in Figure S7). The msd is quadratic in time (msd $\propto t^2$, or distance $\propto t$), a characteristic signature of purely ballistic transport. The EP velocity at this momentum is $39 \pm 1$ µm/ps, 13% of the speed of light. The distinctive interference-like features in the MUPI image profiles arise from the tilted probe plane wave interacting with the polariton population, which we successfully reproduce in Monte-Carlo simulations (Fig. 1e and Supplementary Note 2).

To elucidate optical contrast generation in MUPI, we turn to transient angle-resolved spectroscopy (Fig. 1h, i). The excited state EP dispersion can be monitored any distance away from the excitation to probe how propagating *versus* non-propagating populations affect the dispersion. At the excitation spot, pump-generated excitons lead to a shift of the optical dielectric response of the perovskite semiconductor[39], which results in a uniform blueshift of all LP branches (Fig. 1h, modeled in Supplementary Note 3). This exciton-induced dispersion modulation reinforces that changes to LP spectra do not necessarily reflect EP dynamics[40]. In contrast, the photoinduced dispersion 1.1 µm away from the excitation and at 1 ps pump-probe time delay (Fig. 1i) reflects only the EP population, since excitons are effectively immobile on this timescale. The primary feature in Fig. 1i is that LP branches are broadened compared to ground state branches (Supplementary Note 3). We attribute this broadening to self-energy renormalization as a result of EP–EP interactions[41]. In our nonlinear experiment, these interactions manifest themselves as a blockade-like effect[42], wherein the presence of pump-generated EPs precludes the probe from exciting the LP branch at exactly the same energies and momenta. This nonlinear interaction is momentum- and energy-specific, allowing us to directly correlate MUPI measurements of polariton transport to specific points in the dispersion, as explored below. These EP-EP interactions also generate large contrast, which we leverage to reach exceptional signal-to-noise ratios with few-minute measurements per time delay and pump fluences below 10 µJ/cm$^2$.

## Lattice phonons strongly renormalize EP propagation

We now turn to a detailed analysis of how EP transport is affected by interactions with the material lattice. We leverage the momentum-selectivity of MUPI to directly image the transport of EPs at different parts of the dispersion, i.e., as a function of excitonic character (Figs. S15–S16). By directly tracking EP propagation in the time-domain, we extract an empirical EP velocity. We then compare this velocity to the expected EP group velocity, which we extract from the gradient of the experimental dispersion ($\partial\omega/\partial k$, where $\omega$ is the angular frequency). Our key result is displayed in Fig. 2a, where the data points (open symbols) are the measured velocity, and the solid lines show the expected group velocity for two EP branches. The bottom panel of Fig. 2a shows results of quantum dynamical simulations that we will return to below. When the probe detuning from the exciton energy ($|E-E_{ex}|$) is large, EPs are photon-like, and the experimentally-observed propagation velocity matches the expected group velocity. As the detuning decreases, moving toward more exciton-like species, we find an increasingly large deviation from the expected group velocity.

The observed deviation of the velocity can be rationalized by exciton-mediated EP–matter scattering, an interaction that becomes stronger as the excitonic character of the EP increases[43,44]. We rule out EP–exciton and density-dependent interactions: excitons and other laser-generated species are only present in the sub-micron region defined by the laser excitation spot; yet EPs maintain constant velocities throughout the fitted propagation range, including long after they escape the photoexcited region by several microns (Fig. S7). Thus, EP–matter interactions occur homogeneously in space, suggesting that EP–phonon or EP–defect scattering are responsible for the velocity renormalization. Since exciton–phonon scattering dominates the transport properties of excitons in 2D halide perovskites at room temperature[34,45,46], we infer that exciton-mediated EP–phonon scattering is the dominant contribution to the velocity renormalization. We experimentally verify this hypothesis by performing MUPI experiments at 5 K, which reduces dynamic disorder (phonon scattering) by several orders of magnitude compared to room temperature[47] while negligibly affecting static disorder. EPs at 5 K propagate at the group velocity corresponding to the experimental dispersion (Fig. S17), confirming that the observed velocity renormalization at room temperature results primarily from EP–phonon scattering.

In Fig. 2b, we plot the EP velocity renormalization (the percentage decrease in measured velocity compared to the expected velocity) as a function of excitonic fraction for both room temperature and cryogenic measurements. The data reveals substantial velocity renormalization of up to 40% for half-exciton EPs at room temperature. In contrast, cryogenic measurements display no renormalization for EPs with up to 88% exciton content. Data for two different room-temperature samples with different zero-momentum exciton–cavity detuning are plotted (Figs. S15–S16). For a given exciton fraction, the two samples display nearly identical percentage velocity renormalizations, indicating that the latter is independent of cavity detuning. These results suggest that the velocity renormalization is primarily sensitive to the excitonic fraction, not the EP effective mass or absolute velocity. This observation is consistent with our hypothesis of exciton-mediated EP–lattice scattering. Furthermore, unlike the dispersion renormalization observed at high EP densities in the condensate regime[48], the EP dispersion itself is not renormalized in our structures. This aspect confirms that the velocity reduction we observe is not caused by density-dependent repulsive interactions or particle hybridization, but rather by scattering. Our results suggest that the commonly-used assumption that the gradient of the dispersion ($\partial\omega/\partial k$) corresponds to the polariton velocity may not be accurate for polaritons with high matter character propagating in disordered environments.

To further understand the nature of EP propagation in the presence of exciton-mediated interactions, we fit the MUPI-extracted msd to a power law, msd $\propto t^\alpha$, for EPs with different exciton fraction (Fig. 2c). For exciton fractions below 50%, we observe ballistic transport ($\alpha = 2$) despite the strongly renormalized velocities. This result indicates that coherent propagation is preserved in our systems for EPs with up to 50% exciton character, since ballistic (wavelike) transport implies long-range coherence[49]. Steady-state double-slit interference measurements confirm the spatial coherence of EPs in this system

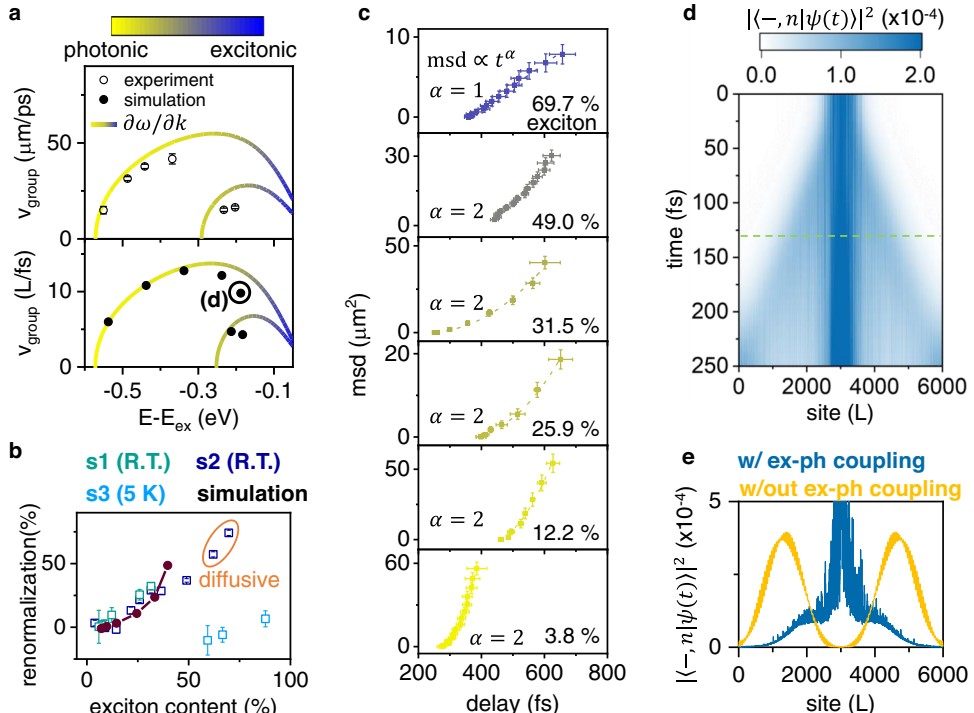

**Fig. 2 | EP propagation and scattering. a** Expected group velocity from the gradient of the dispersion (solid curve) *vs.* measured transport velocity (symbols) for each probing condition, showing an increasing deviation as exciton content is increased. $E_{ex}$ corresponds to the energy of the exciton resonance. The top panel shows results from MUPI experiments at room temperature, while the bottom panel shows results from quantum dynamical simulations. **b** EP velocity renormalization as a function of exciton content for both experimental and simulation results. Experiments are carried out at both room temperature (R.T.) and 5 K. Experiments suggest a transition to diffusive (incoherent) transport above 50% exciton content for R.T. experiments, while simulations suggest decoherence sets in above 40% exciton content. Circled data points are best fit with a diffusive ($\alpha = 1$) model (see panel **c**), for which velocity is not a well-defined quantity; to estimate a percentage renormalization for this data, we fit an effective velocity assuming a ballistic model ($\alpha = 2$). **c** Fits (dashed curves) to the msd extracted from MUPI experiments at R.T. Error bars are one standard deviation. **d** Quantum dynamical simulations of the spreading of the EP probability density $|\langle -, n|\psi(t)\rangle|^2$ as a function of site location (L), starting with a localized initial wave-packet with an energy-window centered at $E-E_{ex} = -0.19 \pm 0.025$ eV, corresponding to the circled symbol in panel (**a**). **e** EP probability density at the horizontal dashed line cut of panel (**d**), for simulations with and without exciton-phonon coupling.

(Fig. S18), and the correspondence between ballistic and coherent transport is further supported by computing the purity of the density matrix of the polariton system (Fig. S13). Nevertheless, above 50% exciton content, we find that the msd for room temperature samples is best fit with $\alpha = 1$, indicating diffusive (incoherent) EP propagation. In contrast, our cryogenic measurements indicate preserved ballistic transport even for EPs with up to 88% excitonic character (Fig. S17). These results show that EP–phonon interactions are not just responsible for velocity renormalization, but also for the transition from ballistic to diffusive transport. Our detailed picture of EP–phonon interactions, uniquely enabled by high-resolution time-domain imaging of EP transport, reveals the rich spectrum of propagation dynamics across both photon-like and exciton-like polaritons: from ballistic and unaffected by phonons, to ballistic but severely slowed by phonon interactions, to diffusive transport. During the lengthy review process of our manuscript[50], a closely-related study performed in organic Bloch surface wave polaritons at room temperature was submitted and published; this study also displayed a group velocity renormalization and a transition from ballistic to diffusive transport[51], suggesting the behavior we are observing is general.

We turn to theory to shed more light on the nature of EP transport in the presence of dynamic disorder. Following past work on charge transport in halide perovskites[52–54], we appeal to a dynamic disorder model where transport in the absence of cavity hybridization occurs purely diffusively via the transient localization mechanism[55,56]. Our simplified, one-dimensional model Hamiltonian describes a single exciton coupled to phonons and an optical cavity (Supplementary Note 4). Coupling to the cavity has a dramatic effect on the transport

properties of EPs[57]. In particular, while the instantaneous eigenstates which govern transport of the pure exciton-phonon system exhibit significant localization, polaritonic eigenstates are largely delocalized. This qualitative change shields photon-like EPs from phonon scattering, and leads to ballistic spreading of the polariton wavepacket (Fig. S14). As the excitonic character of EPs increases, the percentage of localized state character correspondingly increases (non-propagating states in Fig. 2d). Interestingly, however, a substantial part of the EP population continues to propagate ballistically even for large exciton content (cone-like wavepacket in Fig. 2d). Although ballistic propagation is preserved, the wavepacket velocity is substantially reduced by phonon-mediated transient localization, in close agreement with our experimental observations. Figure 2e compares the computed probability distributions at 130 fs for EPs with and without exciton-phonon coupling (setting $\gamma = \alpha = 0$ in Equation S1), confirming that phonon interactions can substantially slow EP wavepackets without destroying their coherence. Our simulations predict that a high degree of coherence is maintained for up to 40% exciton fraction (Fig. S13), in rough agreement with the 50% threshold observed experimentally. Our quantum dynamical simulation results are summarized in Fig. 2a, b (filled symbols), with trends in semi-quantitative agreement with experiments despite the use of a simplified model. This correspondence provides further support to our hypothesis that group velocity renormalization and decoherence are induced by EP–phonon interactions.

## Uncoupled excitons act as reservoir states but not sinks
The above-gap pump excitation we have used thus far generates a large population of long-lived excitons that are not coupled to the

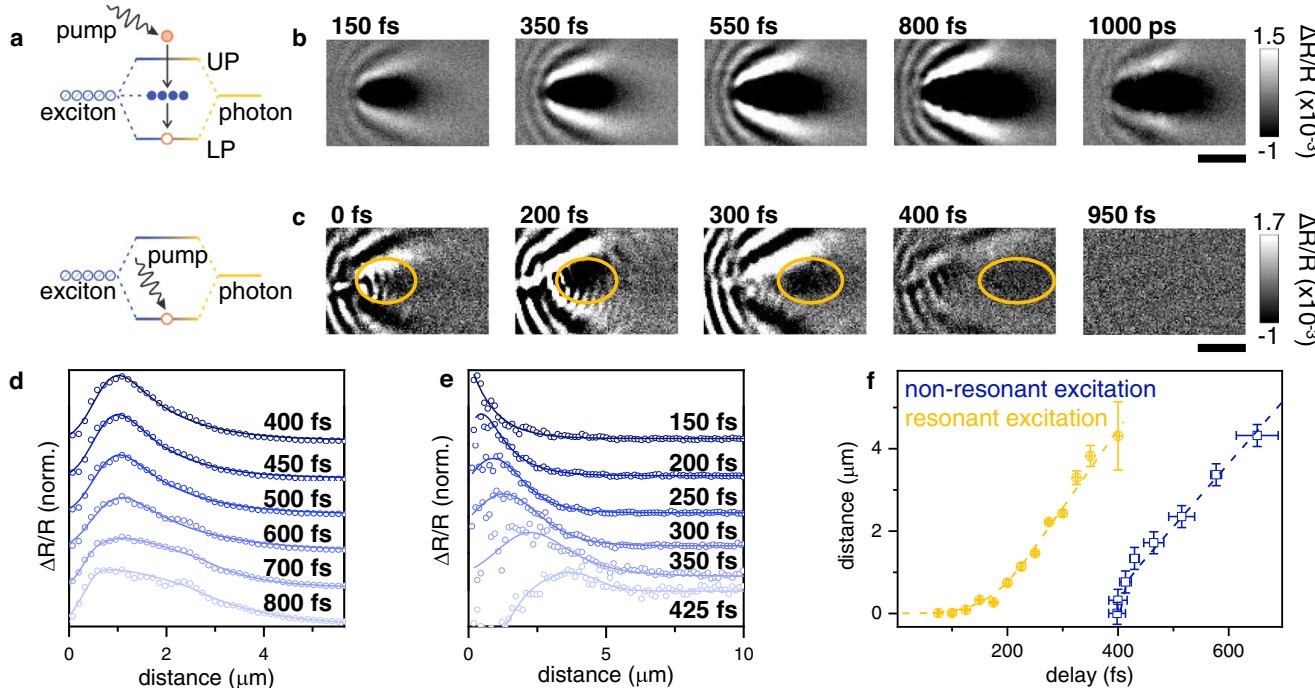

**Fig. 3 | Resonant vs non-resonant excitation of EPs. a** Tavis-Cummings model where hybridization between a cavity mode and N excitons leads to N-1 uncoupled excitons (blue circles, also known as dark states). The two different pump excitation conditions for panels (**b**) and (**c**) are illustrated at top and bottom, respectively. **b** Non-resonantly excited EPs probed at $E = 1.91$ eV and $k = 6.90$ μm$^{-1}$ (LP branch) propagate rapidly in the first ~800 fs and then remain static for a nanosecond. **c** Resonantly excited EPs with both pump and probe at $E = 1.91$ eV and $k = 6.90$ μm$^{-1}$ (LP branch) display a fast-propagating EP wavepacket that disappears after ~800 fs.

The yellow circle highlights the EP wavepacket on top of the strong scattering background; the latter is difficult to avoid in degenerate pump-probe microscopy. Scale bars are 2 μm. **d, e** Evolution of spatial profiles of EPs for non-resonant (**d**) and resonant (**e**) excitation. **f** EP transport extracted from the data in panels (**d**, **e**). Error bars are one standard deviation; see Supplementary Note 5 for detailed analysis. The dashed lines are from Monte-Carlo simulations of EP transport incorporating EP-EP scattering in the case of resonant excitation.

cavity[58,59]. To elucidate the role that these excitons play in EP assemblies, we now compare transport behavior upon non-resonant (above-gap) excitation *versus* resonant excitation of the LP branch, as illustrated in Fig. 3a. Figure 3b displays MUPI data when pumping above-gap and probing the LP branch up to a time delay of 1 ns; surprisingly, EP-associated signals are present 1 ns after photoexcitation, despite the EP lifetime of ~240 fs in our system (Table S1). Such long lifetimes of the LP branch are regularly observed in time-resolved spectroscopy of EP assemblies[58,60], but recent reports cast doubt on the nature of this signal[40]. The observed species propagates over many microns in less than a picosecond, establishing unambiguously that the signal corresponds to EPs. Indeed, the propagation itself ceases after a few hundred femtoseconds (limited by the intrinsic EP lifetime), but the signal persists and remains static from ~800 fs to 1 ns. These observations lend support to the exciton reservoir hypothesis[58,60], wherein uncoupled excitons populated by the non-resonant pump can scatter into the LP branch, continuously refilling the LP population throughout the exciton reservoir lifetime (6 ns in our case, Fig. S5). This hypothesis is further confirmed by resonant excitation data in Fig. 3c, wherein the exciton reservoir is not populated by photoexcitation. Under resonant excitation, we observe ballistic propagation of an EP wavepacket (highlighted with the yellow circle), with the signal disappearing entirely after 800 fs. These results also suggest that population transfer from the LP branch to the exciton reservoir[61] is negligible in our sample.

Figure 3d–f displays the transport properties of non-resonantly vs resonantly-populated EPs (additional data and analysis are presented in Supplementary Note 5). In the long-time limit, both types of EPs propagate at matching velocities (Fig. 3f and S19–S20), indicating that the exciton reservoir populated under non-resonant excitation does not influence the extracted EP velocities. The early-time behavior,

however, is different. For the data displayed in Fig. 3, non-resonant excitation leads to slow EP population buildup reaching a maximum at ~400 fs after photoexcitation; in contrast, resonant excitation leads to full population buildup within 100 fs. In the latter case, we observe slow but accelerating initial propagation; we attribute this result to the high density of EPs at the excitation location which results in strong EP–EP scattering, slowing down the initial EP propagation. As EPs decay and propagate, the EP density decreases and EP propagation speed concomitantly increases, reaching its final transport velocity ~100 fs after the signal appears. We reproduce this acceleration in Monte-Carlo simulations which take into account density-dependent EP–EP scattering (dashed lines in Fig. 3f, see Supplementary Note 2 for details). These temporally inhomogeneous dynamics, similar to those recently discovered in phonon-polaritons[29], further emphasize the importance of tracking EPs throughout their lifetimes to characterize interactions between the many different component excitations of polaritonic systems.

### Generalizability of MUPI

Finally, we show that MUPI is generalizable to many different polaritonic assemblies, though we do not perform a detailed analysis here. Figure 4 displays MUPI snapshots of propagating polaritons at 1 picosecond delay for four different systems: (a) Cavity polaritons in a Distributed Bragg Reflector (DBR) cavity of a layered perovskite, showing long-range transport beyond the 19 μm field of view; (b) Self-hybridized EPs (no artificial cavity) in the same perovskite[32]; (c) Self-hybridized EPs in a transition metal dichalcogenide (WSe$_2$) slab[27,62]; and (d) Plasmon-exciton polaritons (plexcitons) in systems comprised of amorphous 6,13-Bis(triisopropylsilylethynyl)pentacene (TIPS-Pn) deposited on plasmonic silver thin films. We observe ballistic transport on femtosecond scales (except for plexcitons where the full

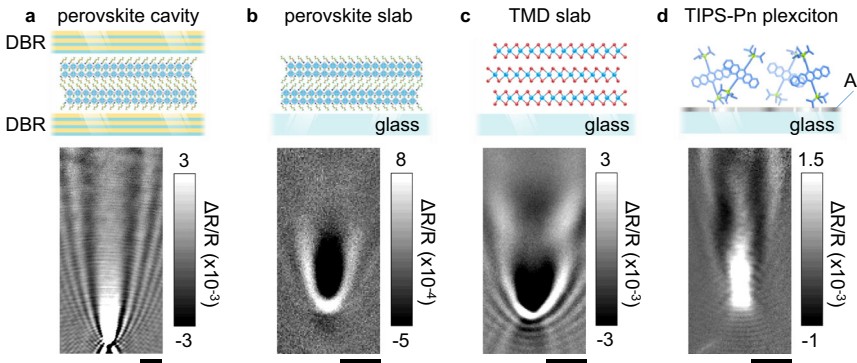

**Fig. 4 | Generalizability of MUPI.** Snapshots of MUPI at 1 ps pump-probe time delay following non-resonant excitation for: **a** EP transport in $CH_3(CH_2)_3NH_3)_2(CH_3NH_3)Pb_2I_7$ flanked by two Distributed Bragg Reflector (DBR) mirrors; **b** EP transport in a 1.13 μm thick layered halide perovskite slab $(CH_3(CH_2)_3NH_3)_2(CH_3NH_3)Pb_2I_7$ with no artificial cavity (Fig. S21a); **c** EP transport in a 69 nm thick flake of $WSe_2$ with no artificial cavity (Fig. S21b); **d** Plexciton transport in a 30 nm Ag film/50 nm TIPS-Pn amorphous film heterostructure. Scale bars are all 2 μm.

propagation occurs within our instrument response), followed by static signals over more than tens of picoseconds when non-resonant excitation is used. As expected, propagation distances in self-hybridized cavities are much shorter compared to artificial cavities due to the shorter EP lifetime. Plexcitons exhibit much longer propagation lengths thanks to the highly dispersive plasmon modes imparting group velocities approaching the speed of light, despite the disordered nature of the amorphous molecular system used here.

In conclusion, we have developed a momentum-selective ultrafast optical imaging approach that directly visualizes EP propagation in real space and time in a wide range of emerging semiconductors. Importantly, we find that the group velocities of EPs with large excitonic character are substantially renormalized through scattering with the material lattice, which we attribute primarily to EP–phonon scattering in our layered halide perovskite microcavities. These results indicate that the commonly used assumption that the polariton velocity corresponds to the gradient of the dispersion may not be a good approximation, particularly for highly excitonic polaritons. Remarkably, however, EPs maintain ballistic transport even in these strongly-interacting environments for up to half-exciton EPs. For EPs with excitonic fractions higher than 50%, we observe diffusive, incoherent transport at room temperature. Quantum dynamical simulations and cryogenic measurements indicate that the transition from non-interacting, to ballistic with renormalized velocities, to incoherent transport arises from the interplay of transient localization of EPs induced by strong exciton–phonon interactions and partial shielding of these interactions by hybridization with the cavity. Overall, we have established a general framework that enables precisely balancing EP coherence, velocity, and nonlinear interactions for any given polaritonic architecture. We believe these measurements will be crucial to optimize next-generation polaritonic technologies that seek to truly harness the best of their light and matter components, for example to incorporate single-photon gates[11,12,14] into scalable quantum circuits requiring long-range propagation.

## Methods

Details of sample synthesis, cavity fabrication, optical measurements and theory are provided in the Supplementary Information.

A schematic of MUPI is shown in Fig. S1. A 40 W Yb:KGW ultrafast regenerative amplifier (Light Conversion Carbide, 40 W, 1030 nm fundamental, 1 MHz repetition rate) seeds an optical parametric amplifier (OPA, Light Conversion, Orpheus-F) with a signal tuning range of 640–940 nm and an average pulsewidth of 60 fs. For non-resonant excitation experiments, the second harmonic of the fundamental (515 nm) is used as a pump pulse, and the OPA signal is used as probe. For resonant excitation (single-color)

experiments, the OPA signal is split in pump and probe beams. Group delay dispersion is partially pre-compensated using a pair of chirped mirrors (Venteon DCM7). The pump pulse is sent collimated into a high numerical-aperture objective (Leica HC Plan Apo 63x, 1.4 NA oil immersion), resulting in diffraction-limited excitation on the sample. Typical pump fluence incident on the semiconductor is 5 μJ/cm². The probe is sent to a computer-controlled mechanical delay line for control over pump-probe time delay, and is combined with the pump beam through a dichroic mirror. An $f = 250$ mm widefield lens is inserted prior to the dichroic mirror to focus the probe in the back focal plane of the objective for widefield illumination of the sample. A tilting mirror placed one focal length upstream of the widefield lens allows tuning the angle at which the widefield probe illuminates the sample, thus allowing probing at any momentum up to a maximum of $k/k_0 = 1.4$, limited by the numerical aperture of the objective.

Backscattered light from the sample is collected through the objective, directing the light to two different detection paths. For angle-resolved linear and transient reflectance (Fig. 1 h–j), the back focal plane of the objective is projected on the entrance slit of a home-built prism spectrometer using a pair of lenses ($f_1 = 300$ mm and $f_2 = 100$ mm). For real-space MUPI imaging, this projected back focal plane image is Fourier transformed again into real-space using a 150 mm lens, forming an image on a CMOS camera (Blackfly S USB3, BFS-U3-28S5M-C). Both the spectrometer camera and the real-space camera are triggered at double the pump modulation rate, allowing the consecutive acquisition of images with the pump ON followed by the pump OFF.

## Data availability

All raw data are displayed in Figs. 1–4 of the main text and Figs. S2–S21 of the Supplementary Information. Raw image files are available from the corresponding authors upon request.

## Code availability

The source code for quantum dynamics simulations is available at[63] https://github.com/arkajitmandal/BalisticPolaritons.

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

## Acknowledgements

This material is based upon work supported by the National Science Foundation under Grant Numbers DMR-2115625 (M.D.) and CHE-1954791 (D.R.R.). Revisions were primarily supported by the National Science Foundation under Grant Number CHE-2203844 (M.D.). M.D. also acknowledges support from the Arnold and Mabel Beckman Foundation through a Beckman Young Investigator award. Synthesis of WSe$_2$ (S.L.) was supported by the NSF MRSEC program through Columbia in the Center for Precision-Assembled Quantum Materials (DMR-2011738). This work used the Extreme Science and Engineering Discovery Environment (XSEDE), which is supported by National Science Foundation grant number ACI-1548562 (allocations: TG-CHE210085). Specifically, it used the services provided by the OSG Consortium, which is supported by the National Science Foundation awards #2030508 and #1836650.

## Author contributions

D.X. and M.D. conceived and designed the experiments. D.X. and J.B. developed the instrument. D.X., S.C., I.L. and H.S. acquired the experimental data. S.L. synthesized WSe2 crystals. D.X. analyzed the data. A.M. and D.R.R. developed and performed the quantum dynamical simulations. D.X., A.M., D.R.R. and M.D. wrote the manuscript, with input from all authors.

## Competing interests

The authors declare no competing interests.
