## [Peer Review File · Nature Communications]

REVIEWERS' COMMENTS

Reviewer #1 (Remarks to the Author):

I have read the response to my comments and believe that the manuscript is suitable for publication in Nature Communications in its current form.

Regarding the response: I agree with the authors that the strongest contributor to the signal away from the pump under their specific conditions is the resonant χ -3 interaction with propagating EPs. My point was that under different conditions, one may have contributions from other species dominate and/or contribute to the signal. However the authors raise a fair point that k-space asymmetry in the pump-probe signal can help distinguish this.

I have checked and the authors are correct that there is no mention of diffusive transport in 10.1021/acsp Photonics.7b01332. If anything, those results are poorly interpreted. However given that polariton lifetimes in that work are 10 fs and the corresponding group velocities, it seemed clear (at least to me) from the continued spreading at 10 ps (and the spatial extent) that transport could only be diffusive.

I was not aware of the recent work by Schwartz et al. It is probably very little comfort to the authors, but after reading that paper, I do not see the level of novelty one would expect at the level for Nature Materials either. There are no surprises in that work (and less in methods development than in the current paper).

Given the similarities with this manuscript, I agree with the authors that any further delays should be avoided.

Reviewer #1 (Remarks to the Author):

I have read the response to my comments and believe that the manuscript is suitable for publication in Nature Communications in its current form.

Regarding the response: I agree with the authors that the strongest contributor to the signal away from the pump under their specific conditions is the resonant χ -3 interaction with propagating EPs. My point was that under different conditions, one may have contributions from other species dominate and/or contribute to the signal. However the authors raise a fair point that k-space asymmetry in the pump-probe signal can help distinguish this.

I have checked and the authors are correct that there is no mention of diffusive transport in 10.1021/acsphotonics.7b01332. If anything, those results are poorly interpreted. However given that polariton lifetimes in that work are 10 fs and the corresponding group velocities, it seemed clear (at least to me) from the continued spreading at 10 ps (and the spatial extent) that transport could only be diffusive.

I was not aware of the recent work by Schwartz et al. It is probably very little comfort to the authors, but after reading that paper, I do not see the level of novelty one would expect at the level for Nature Materials either. There are no surprises in that work (and less in methods development than in the current paper).

Given the similarities with this manuscript, I agree with the authors that any further delays should be avoided.

Author response: We thank Reviewer 1 for their recommendation to publish our work.